# IL-18 Binding Protein, a biomarker of strength maintenance after surgery but reduced physical performance in age-related sarcopenia

**Richard Paul[1,2], Christos Rossios[1], Aaron C. Hinken[3], David Neil[3], Alan Russell[3,4], Miles D. Witham[5,6], Mark J. Griffiths[1,7], Paul R. Kemp[1]***

**1** Cardiovascular and Respiratory Interface Section, National Heart and Lung Institute, VPD Building Hammersmith Campus, Imperial College London, London, United Kingdom, **2** Department of Intensive Care, Guy's and St. Thomas' NHS Foundation Trust, London, United Kingdom, **3** Muscle Metabolism Discovery Performance Unit, GlaxoSmithKline, Inc., Collegeville, Pennsylvania, United States of America, **4** Edgewise Therapeutics, Boulder, Colorado, United States of America, **5** AGE Research Group, Translational and Clinical Research Institute, Faculty of Medical Sciences, Newcastle University, Newcastle, United Kingdom, **6** NIHR Newcastle Biomedical Research Centre, Newcastle upon Tyne Hospitals NHS Foundation Trust, Cumbria, Northumberland, Tyne and Wear NHS Foundation Trust and Newcastle University, Newcastle, United Kingdom, **7** Peri-operative Medicine, Barts Heart Centre, St Bartholomew's Hospital, London, United Kingdom

* p.kemp@imperial.ac.uk

## Abstract

Inflammation is thought to contribute to muscle loss in acute and chronic sarcopenia. Which inflammatory proteins contribute to sarcopenia in any condition is not clear. In a well-characterised cohort of patients experiencing acute sarcopenia following surgery, we used a proteomic screen of plasma to identify proteins associated with the change in strength. We compared change in handgrip strength over 7 days in surgery patients with plasma protein levels quantified by SOMAscan before and 24h after surgery. Surgery increased circulating concentrations of 295 proteins and decreased 301. Analysis of the day 1 protein levels showed that IL-18BP associated with maintenance of strength. To further investigate relationships between IL18BP and strength, IL-18BP as well as its ligands IL-18 and IL-37, were quantified by ELISA and in surgery patients and in 129 individuals (68 women) with age-related sarcopenia recruited to the Leucine and/or ACE inhibitor (LACE) trial. In LACE participants, the proteins were compared to grip strength, quadriceps maximal voluntary contraction (QMVC) and 6-minute walk distance (6MWD) and baseline SARC-F score. In the LACE cohort, IL-18BP was negatively associated with grip strength in men but not women, at baseline (r = −0.314, p = 0.014) and 12 months (r = −0.446, p = 0.001). QMVC and 6MWD showed similar associations. IL-18BP was associated with SARC-F in men (r = 0.389, p = 0.003) but not women. Investigation of SOMAscan data from surgery patients at baseline showed similar inverse associations of IL-18BP with strength. Comparison of circulating IL-18BP with the muscle transcriptome in these patients showed negative enrichment for mitochondrial genes. Analysis of the ligands

**Data availability statement:** All relevant data are within the paper and its Supporting information files.

**Funding:** The surgery cohort study was funded by the NIHR Respiratory Diseases Biomedical Research Unit at the Royal Brompton and Harefield NHS Trust (BRU 6279) which funded RP with MG and PK as applicants. The LACE trial (project reference 13/53/03) was funded by the Efficacy and Mechanism Evaluation (EME) Programme, an MRC and NIHR partnership (https://www.nihr.ac.uk/explore-nihr/funding-programmes/efficacy-and-mechanism-evaluation.htm) and funded CR. The Principal Investigator of the award was MW with PK, as a co applicant. These funders had no role in study design, data collection and analysis, decision to publish, or preparation of the manuscript. The SOMAscan analysis was funded by a GSK award (40453995) to PK with AR, AH and DN contributing to the design of this aspect of the work.

**Competing interests:** The Authors have declared that no competing interests exist.

showed that free IL-18 was proportional to 6MWD. After surgery high IL-18BP levels associate with maintenance of strength but circulating IL-18BP concentrations are associated with reduced muscle strength in men with sarcopenia. These data are consistent with known effects of IL-18BP ligands on the maintenance of mitochondrial function.

## Introduction

Loss of muscle mass and strength commonly accompany many chronic and acute diseases, as well as old age – the syndrome of sarcopenia [1]. This loss of muscle mass and strength can lead to a reduction in the ability to perform the tasks of daily living and a consequent requirement for care, as well as falls, injury, prolonged recovery from illness, reduced quality of life and shorter life expectancy [2]. Muscle strength is a product of several factors including the amount of muscle tissue, its type and composition, and energy provision. Mechanisms that influence these factors contribute to muscle strength; different mechanisms may have separate and independent effects on each of these factors contributing to overall strength.

Inflammation is a common factor in a range of both chronic and acute conditions associated with muscle loss. Cytokine mediators of inflammation are known regulators of muscle mass and strength [3]. Changes in muscle mass are the product of an altered balance in protein turnover the product of reduced synthesis and/or increased breakdown. It is well known that inflammatory stimuli (e.g., infection, tissue damage or dysfunction) can lead to muscle breakdown and that the stimulation of muscle proteolysis can occur in response to pro-inflammatory cytokines (e.g. TNFα, IL-6, IFNγ) [4–8]. This increase in protein breakdown is likely required to provide amino acids and energy for tissue repair and recovery [9,10].

Although inflammation contributes to muscle wasting, it is also a key component of muscle growth and regeneration via the clearing of debris and activation of satellite cells. For example, IL-6 increases the expression of the atrophy promoting ubiquitin ligase MuRF1 [11] and promotes muscle loss following infusion [12], whereas IL-6 deficiency inhibits muscle loss in sepsis [13]. However, IL6 is also increased in response to exercise and contributes to hypertrophy as IL-6$^{-/-}$ mice have a reduced hypertrophic response to overload with reduced myonuclear accretion [14].

Inflammatory cytokines are also noted regulators of energy metabolism providing another potential mechanism to affect muscle strength. IL-15 activity affects muscle mass and strength by modulating the production of mitochondria [15,16]. IL-18 also appears to contribute to the regulation of energy and oxidative metabolism under basal conditions as IL-18$^{-/-}$ and IL-18RA$^{-/-}$ mice have ectopic lipid disposition and impaired AMPK signalling in the muscle together with impaired glucose tolerance [17] raising the possibility that it too could increase physical performance. Consistent with this observation IL-18 activates AMPK in myotubes and increases fat oxidation [17]. As well as effects in muscle, IL-18 limits adipose tissue expansion [18] an effect likely to contribute to impaired glucose tolerance.

The activity of inflammatory cytokines can be modulated by circulating binding proteins and soluble receptors. For example, IL-15 can be bound and neutralised by a soluble form of its receptor, IL-15RA, and levels of this receptor in the circulation of patients undergoing surgery are associated inversely with markers of mitochondrial dysfunction in patients about to undergo aortic surgery [19]. Similarly, IL-18 binding protein (IL-18BP) binds and neutralises the activity of IL-18, a pro-inflammatory cytokine that is thought to be a major contributor to sepsis, a condition associated with a loss of muscle mass and strength. In this scenario, IL-18BP expression is elevated by interferon-γ as part of the inflammatory response and limits the effects of IL-18 reducing the likelihood of macrophage activation syndrome a potentially fatal over-stimulation of the immune response [20].

Given the contribution of inflammatory cytokines to muscle degradation, the response to exercise and energy provision, it seems likely that the particular response to any individual inflammatory cytokine will be dependent on the level to which the cytokine is raised, the duration of elevation and the presence of other inflammatory cytokines, and regulators of those cytokines. Furthermore, given the pleiotropic effects of inflammatory cytokines the effects of the cytokines on performance may be indirect (e.g., by affecting energy supply).

Evidence supporting roles for IL-18 in both the loss of muscle and physical performance and the regulation of oxidative metabolism have been found in humans. For example, a polymorphism that associates with lower IL-18 levels was associated with better physical performance in older individuals [21]. In this study, a one standard deviation increase in serum IL-18 was associated with being in the 20% of slowest walkers and those with poorer short physical performance battery (SPPB) scores. Other studies and a meta-analysis have not fully supported this finding. Higher plasma IL-18 concentrations were also associated with sarcopenia in a population of older people [22]. Similarly, IL-18 is elevated in obesity, where it associates with reduced physical performance. In support of a role for IL-18 in regulating metabolism and promoting lipid oxidation, low IL-18 levels were associated with muscle fat accumulation in patients infected with HIV [23]. However, it is difficult to draw firm conclusions, as the role of IL-18BP in these conditions is rarely analysed, but given its high binding affinity for IL-18 and the molar excess of IL-18BP compared to IL-18 leading to very tight regulation of free (active) IL-18 levels, IL-18BP levels are likely to be critical. To date we are unaware of any analysis of IL-18BP and muscle mass, strength or metabolism.

In this study, we aimed to understand the loss of muscle strength in response to cardiac surgery. We undertook a two-stage investigation, firstly using a proteomics-based approach to identify inflammatory mediators that increased in a cohort of patients undergoing aortic surgery, and secondly undertaking detailed cross-sectional and longitudinal studies in two contrasting populations – an expanded cohort of acute sarcopenia patients and another with chronic sarcopenia of a promising candidate inflammatory marker – IL-18BP. From the proteomic analysis, we hypothesised that elevated IL-18BP would reduce the pro-inflammatory effects of IL-18 and that this would lead to positive association of IL-18BP with muscle mass or strength in individuals with sarcopenia.

## Materials and methods

### Patients

The samples and physiological data used in this study are from two separate clinical studies the main trial results of which have been reported elsewhere. The first study was an analysis of the acute response of muscle loss in patients undergoing aortic surgery [24] and the second was a randomised clinical trial of Leucine and an Angiotensin Converting Enzyme (ACE) inhibitor in sarcopenic individuals, the LACE study [25].

**Acute sarcopenia cohort.** Forty-one patients undergoing elective aortic valve surgery at the Royal Brompton Hospital were recruited as previously described (National Research Ethics Committee 07/Q0204/68) [26]. Participants gave written and informed consent and were recruited between November 2013 and August 2015 (trial registration NCT03354767). Rectus femoris cross-sectional area ($RF_{CSA}$) was measured by ultrasound (CX50, Philips Healthcare, Guildford, UK) between day 0 and day 7 after surgery, as previously described [24]. Handgrip and quadriceps strength assessments

with dynamometers (respectively Jamar® hydraulic hand dynamometer, JA Preston Corporation, Clifton, NJ, USA and Lafayette Manual Muscle Tester, MMT, device 01163, Lafayette Instruments Europe Ltd, Leicestershire, UK) were carried out within 1 week prior to surgery and 7 days after surgery, as previously described [26].

**Sarcopenia cohort.**  The samples from the sarcopenia study come from the LACE randomised controlled trial of leucine and perindopril (an ACE inhibitor) (trial registration ISRCTN90094835). Recruitment to the trial and the main outcomes from the trial are described in [25]. Participants from 14 UK centres between April 2016 and December 2019 were eligible for inclusion if they were aged 70 or over with sarcopenia according to the European working group on sarcopenia in older people (EWGSOP) definition (2010) [27]. The trial was approved by the East of Scotland NHS research ethics committee (approval 14/ES/1099) and the UK Medicines and Healthcare Regulatory Authority (EudraCT number 2014-003455-61; Clinical Trial Authorisation number 36888/0001/001–0001). All participants gave written informed consent. Appendicular muscle mass was determined by dual-energy X-ray absorptiometry (DXA) scan, and physical performance at baseline, 6 and 12 months was measured using hand grip dynamometry (Jamar), quadriceps handheld dynamometry (Lafayette) and 6-minute walk distance (6MWD) as described in [27]. SARC-F was recorded as part of the initial screening for sarcopenia with a minimum score of 3 used to take individuals through to randomisation.

## Ethical statement

All participants gave their written informed consent prior to inclusion in the study and Ethical approval was obtained from the Research Ethics Committees identified above. The studies were carried out in accordance with the ethical standards laid down in the 1964 Declaration of Helsinki and its later amendments.

## Blood samples and analysis

For the acute sarcopenia cohort, plasma was obtained from blood samples taken prior to the induction of anaesthesia on the day of surgery (pre-surgery) and 24h later (post-surgery), and 7 after surgery. Blood was collected into EDTA tubes, centrifuged to prepare plasma and stored at −80°C until required.

Samples selected for SOMAscan analysis came from male patients with complete functional data and sufficient available sample on pre- and post-surgery (day 1) for SOMAscan analysis, then a further level of filtering to select the 15 patients with the most loss of $RF_{CSA}$ and 15 with the least loss. For the enzyme linked immunosorbent assay (ELISA) analysis of IL-18BP sufficient sample was available from 30 patients on day 0 (4F and 26 M with 23 giving both SOMAscan and ELISA data) and 40 patients on day 1 (10F and 30M with 29 giving both SOMAscan and ELISA data). Baseline details for the full cohort have been published and those for the SOMAscan cohort are given in Table S1 in S1 File.

For the LACE cohort plasma was prepared from blood samples taken into EDTA tubes at the baseline visit (prior to randomisation and treatment allocation) and 6 and 12 months later and stored at −80°C until required. Drop out from the study over the course of the 12-month study has been described previously [25] with 129 individuals providing usable data at baseline (68F, 61M), 103 individuals with usable data at 6 months (53F and 50M), and 86 individuals with usable data at 12 months (43F, 43M). The baseline details for this cohort are in Table S2 in S1 File.

## SOMAscan assay

The plasma samples were sent to Somalogic (Boulder, Colorado, USA) for the SOMAscan® assay of 1300 proteins. All samples passed the appropriate quality control measurements as performed by Somalogic. Data from SOMAscan are provided in Table S3 in S2 File.

## ELISA analysis

Samples were analysed for IL-18BP, IL-18 and IL-37 using Duoset ELISAs (R & D Systems, Minneapolis, MN, USA: respectively DY119, DY318, and DY1975), according to the manufacturer's instructions. Briefly, for each ELISA, 96 well

plates (ThermoFischer) were coated with the first antibody provided in the kit overnight at room temperature, before being washed 3 times with wash solution (PBS tween 0.05%) and blocked in PBS BSA 1% incubated for 2h with plasma diluted to bring samples into the working range for the assay (IL-18BP; 1:3, IL-18; 1:2 and IL-37; 1:2). The antibody additions, incubation times, washings and colouration steps were performed according to the manufacturer's instructions. The colour reaction was terminated with stop solution and the absorbance measured using a Tecan SPARK plate reader at 450nm. Data from the ELISA assays are given in Table S4 in S2 File (surgery cohort) and Table S5 in S2 File (LACE cohort)

### RNAseq analysis

RNA was extracted from frozen quadriceps muscle biopsies taken at the time of surgery (after induction of anaesthesia but before surgery) using TRizol as previously described [28]. RNAseq was performed at the Genome Centre, Queen Mary University of London and all samples and generated libraries passed the QC metrics used by the Genome Centre and the data are provided as RPKM in Table S6 in S2 File.

### Statistical analysis

SOMAscan data were analysed using Metaboanalyst 5.0. Data were $log_{10}$ normalised and autoscaled. Difference between pre-surgery and post-surgery protein levels were calculated using LIMMA in the Metaboanalyst package with multiple testing correction using false discovery rate (FDR) with an FDR of 0.05 taken for statistical significance. To identify associations with loss of strength, SOMAscan data were correlated with loss of quadriceps strength using the bi-weight mid correlation in the R package WGCNA with multiple testing correction using FDR.

Correlations of ELISA measured IL-18BP, IL-18 and IL-37 with physiological parameters were made using Spearman's rho and the significance of differences were calculated by Mann-Whitney U test in Aabel 3.0. As this is an exploratory analysis a p value of 0.05 was taken as significant and no corrections were made for multiple testing.

In graphs where correlations were made using Pearson or bi-weight mid correlation, trend lines are shown. Trend lines are not shown for Spearman's correlations to avoid the implication of linear associations.

RNAseq data: The data set was trimmed to genes that were detected in all samples leaving 15071 transcripts. RPKM were log2+1 transformed as recommended in the WGCNA R package and loaded into the package with the IL-18BP levels from SOMAscan. Bi-weight mid correlation was used to compute correlation coefficients for each gene, and these values were used as a ranking metric for gene set enrichment analysis in the package GSEA with the Hallmark gene sets. GSEA compares the order of genes from pre-defined gene groups (gene sets) in the ranked list with the order found in randomised lists (we used the standard 1000 iterations) and computes the relative enrichment at either end of the list for that group and the likelihood of that enrichment occurring by chance [29].

RNAseq and SOMAscan data are provided in supplementary file.

### Calculation of free IL-18

Free IL-18 was calculated using the law of mass action with a 1:1 association of IL-18 with IL-18BP and a dissociation constant of 40pM [30].

### Results

We hypothesised that the inflammatory proteins most likely to contribute to the loss of strength following surgery would be those that changed in response to surgery. Comparison of pre-surgery plasma protein levels (Table S3 in S2 File) with post-surgery plasma protein levels identified 295 proteins that increased and 301 proteins that decreased (Fig S1 in S4 File, Table S7 in S3 File). To identify proteins that associated with loss of strength, the post-surgery circulating level of proteins that increased following surgery were compared with change in strength (% hand grip strength after 7 days) (Table 1). Of the proteins that increased in expression after surgery, 49 had a positive association with loss of hand grip

**Table 1. Bi-weight mid correlation for proteins with loss of handgrip strength showing the top 10 most correlated proteins.**

| Protein | Biweight mid corr with Hand grip % loss | p value | FDR |
|---|---|---|---|
| STC1 | 0.746 | 2.24E-06 | 0.0010 |
| IL18BP | −0.739 | 3.05E-06 | 0.0010 |
| PTPN11 | 0.713 | 9.61E-06 | 0.0022 |
| FGF23 | 0.686 | 2.88E-05 | 0.0049 |
| EPS15L1 | 0.676 | 4.13E-05 | 0.0056 |
| LIFR | −0.654 | 8.96E-05 | 0.0095 |
| FLRT2 | −0.651 | 9.77E-05 | 0.0095 |
| CEBPB | −0.641 | 1.35E-04 | 0.0115 |
| CNTFR | −0.631 | 1.86E-04 | 0.0141 |
| SERPINE1 | 0.625 | 0.000225 | 0.0153 |

Biweight-mid correlation was calculated using the bicor function in WGCNA with outliers set at 10% comparing day 1 protein level with % gip strength loss over 7 days.

FDR = false discovery rate.

strength whereas 67 had a negative association. Both positively and negatively associated proteins came from a range of different protein groups and included both intracellular and extracellular proteins. The associations of intracellular proteins with change in muscle strength are difficult to rationalise but are likely to reflect tissue breakdown. Of more interest therefore are known signalling proteins and their regulators. The SOMAscan assay showed a reduction in myostatin following surgery, as well as an increase in GDF-15 both of which were consistent with our previous ELISA based analysis [31]. Of these, post-surgery myostatin had a negative association with loss of strength (patients with higher myostatin levels post-surgery lost least strength by day 7, Table S7 in S3 File) whereas post-surgery GDF-15 had a positive association with loss of strength (patients with higher GDF-15 levels on post-surgery lost most strength by day 7, Table S7 in S3 File). Other changes of potential interest include the increase in 3 IGF binding proteins (IGFBP1, IGFBP4 and IGFBP5) all of which also positively associate with loss of strength consistent with a role for IGF1 in the maintenance of strength even though IGF1 itself did not change significantly following surgery in this study.

The protein whose circulating post-surgery levels most positively associated with loss of hand grip strength was Stanniocalcin-1 (STC1), and the most negatively associated with loss of strength (lowest post-surgery levels associated with greatest loss of strength) was IL-18BP (Fig 1). To confirm this association, IL-18BP was quantified by ELISA in a slightly larger cohort including the same samples. We tested 2 different STC1 ELISAs but did not get consistent responses. No further analysis of STC1 was performed. However, there was an association of the SOMAscan measurements with the IL-18BP ELISA (r = 0.84, p < 0.001, Fig S2A in S4 File) and the ELISA analysis confirmed the increase in IL-18BP following surgery (Fig S2B in S4 File) suggesting that the protein measured by SOMAscan was indeed IL-18BP.

IL-18BP was not the only interleukin, interleukin receptor or interleukin binding protein in the day 1 SOMAscan set to associate with change in grip-strength at an FDR q < 0.05. The others were IL-10RA (r = −0.564, p = 0.001, q = 0.017), IL-18RAP (r = −0.568, p = 0.001, q = 0.017), IL-37 (r = −0.532, p = 0.002, q = 0.022 though we were subsequently unable to verify this analysis by ELISA), IL-6ST r = −0.5, p = 0.004, q = 0.028), IL-7 (r = −0.481, p = 0.007, q = 0.034), IL-4R (r = −0.471, p = 0.009, q = 0.039), IL-2RG, r = 0.463, p = 0.01, q = 0.04), IL-2(r = −0.455, p = 0.011, q = 0.04, IL-20 (r = 0.452, p = 0.012, q = 0.046).

The above data suggested that higher levels of IL-18BP following surgery associate with the maintenance of strength following surgery. Therefore, to determine whether the same would be true in patients with clinically relevant, chronic sarcopenia, we quantified IL-18BP in the plasma of patients enrolled in the LACE clinical trial by ELISA and compared it

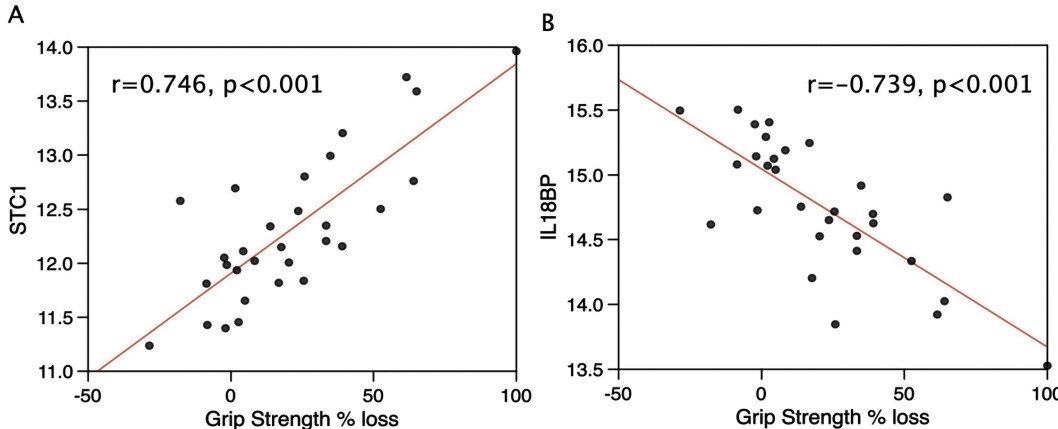

**Fig 1. Associations of STC1, and IL-18BP with loss of grip strength in the acute sarcopenia cohort.** Day 1 plasma levels of the proteins quantified by SOMAscan were correlated using bi-weight mid correlation with loss of grip strength between day 0 and day 7 in the r package WGCNA. STC1 **(A)**, was positively associated with loss of strength, whereas IL-18BP **(B)**, was negatively associated with loss of strength. R values are bi-weight mid correlations.

with physical performance at baseline, and after 6 and 12 months of the trial. Given the difference in muscle mass and strength between men and women in the cohort, the genders were analysed separately. In men, at baseline, IL-18BP was inversely associated with both grip strength (r=−0.314, p=0.014 Fig 2A), quadriceps maximum voluntary contraction (QMVC, r=−0.335, p=0.014, Fig 2B) and six-minute walk distance (6MWD, r=−0.359, p=0.005, Fig 2C). Similar associations of IL-18BP with performance were seen in men at 6 months [IL-18BP vs grip strength (r=−0.442, p=0.003), QMVC (r=−0.330, p=0.025) and 6MWD (r=−0.447, p=0.003) (Fig 2D–2F)] and at 12 months [IL-18BP vs grip strength (r=−0.403, p=0.007) and 6MWD (r=−0.404, p=0.005) (Table 2)]. In women, there were no associations between IL-18BP and physical performance at baseline but there was a negative association with QMVC at 6 (r=−0.442, p=0.003) and 12 months (r=−0.355, p=0.038, Table 2). There was no association of IL-18BP with muscle mass for either gender. Dividing the men into those with plasma IL-18BP above and below median (8.48ng/mL) showed that individuals with higher IL-18BP had a higher SARC-F score (above median 4, (3,6) vs below median 3 (3,4) p=0.003), lower QMVC (above median 12.4kg (17.5, 8.2) vs below median 18.1kg (14.5, 21.6) p=0.004), and lower 6MWD (above median 282m (162, 364) vs below median 370m (266, 413) p=0.025) than those with plasma IL-18BP below 8.48ng/mL (Fig 3). Although median grip strength was lower in those with plasma IL-18BP above 8.48ng/mL this difference did not reach statistical significance (above median 22.8kg (20.0, 24.6) vs below median 23.6kg (20.0, 28.4) p=0.067, Fig 3B). However, baseline IL-18BP did not associate with change in muscle strength or 6MWD performance over 12 months.

These associations of physical performance in the sarcopenia cohort were opposite to those expected from the association seen in patients following surgery. We therefore compared the baseline IL-18BP with the available baseline measurements of physical performance in the acute sarcopenia cohort. This analysis showed that consistent with the analysis of the sarcopenia cohort, in men IL-18BP negatively associated with QMVC (Fig 4), there were too few women in this study to support analysis.

RNAseq data (Table S6 in S2 File) were available from quadriceps biopsies for 14 of the patients for whom we had protein data (all men). To understand the biochemical basis for the association of IL-18BP with reduced physical performance at baseline, we compared circulating IL-18BP with the muscle transcriptome and used gene set enrichment analysis (GSEA) using the 'Hallmark' gene sets to identify pathways that could account for the associations. This analysis showed that in the pre-surgery samples, mRNAs that associated with circulating IL-18BP were negatively enriched for those

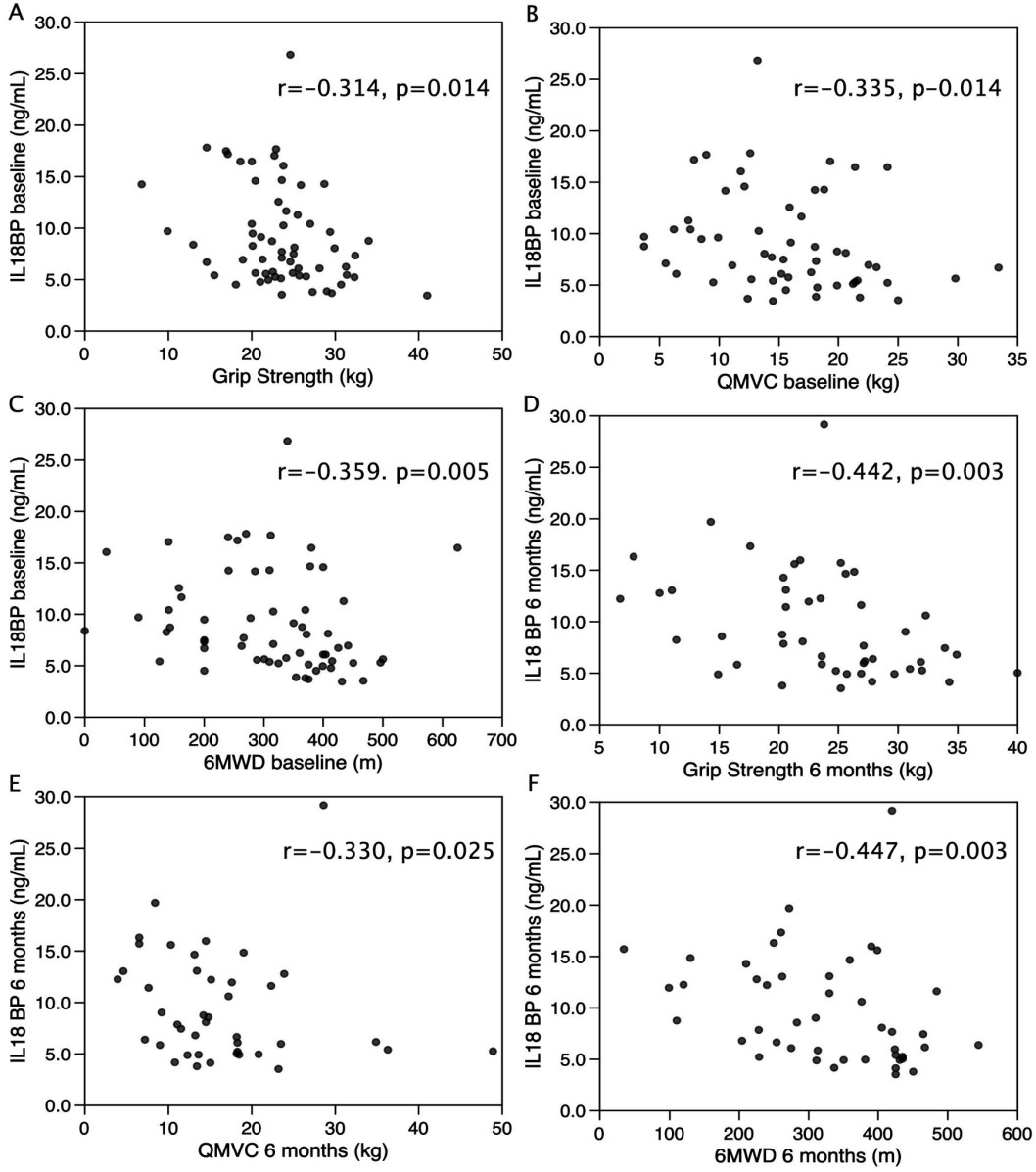

**Fig 2. IL-18BP is inversely associated with physical performance in sarcopenic men recruited to the LACE trial.** IL-18BP was quantified in plasma from men recruited to the LACE trial at baseline and 6 months later and compared with grip-strength **(A, D)**, QMVC **(B, E)** and 6MWD **(C, F)** at the same time point. At both timepoints there was a negative correlation between circulating IL-18BP and physical performance. (Correlations carried out as Spearman's rank correlation).

associated with oxidative phosphorylation (with no other gene sets showing enrichment at an FDR-q of p<0.05, Table 3). GSEA using the GO: Biological Processes gene sets also identified multiple components of mitochondria as negatively enriched with multiple aspects of mitochondrial biology showing a greater than 2-fold enrichment (Table S8 in S1 File). To confirm this analysis, we compared the genes that had a negative correlation coefficient with IL-18BP that was significant at a nominal p value of 0.05 (n=435) against the GO cell compartment division using DAVID functional annotation and also found over-representation of genes with a mitochondrial localisation (57 genes, p=3.6 x $10^{-6}$, Benjamini corrected

**Table 2. Correlation of IL18BP with physical performance in sarcopenic individuals.**

| | | Women | | Men | |
|---|---|---|---|---|---|
| | Parameter | $r_s$ | p | $r_s$ | p |
| **Baseline parameter vs baseline IL18BP (68F/59M)** | **Grip strength** | 0.060 | 0.624 | −0.314 | 0.014 |
| | **Grip strength/arm muscle mass** | 0.200 | 0.103 | −0.357 | 0.006 |
| | **6MWD** | 0.014 | 0.908 | −0.359 | 0.005 |
| | **QMVC** | −0.042 | 0.736 | −0.335 | 0.014 |
| | **SARCF** | 0.062 | 0.610 | 0.389 | 0.003 |
| **6 months parameter vs 6 months IL18BP (42F/47M)** | **Grip strength** | 0.149 | 0.301 | −0.442 | 0.003 |
| | **6MWD** | 0.062 | 0.686 | −0.447 | 0.003 |
| | **QMVC** | −0.442 | 0.003 | −0.330 | 0.025 |
| **12 months parameter vs 12 months IL18BP (37F/41M)** | **Grip strength** | −0.001 | 0.966 | −0.403 | 0.007 |
| | **6MWD** | −0.109 | 0.514 | −0.404 | 0.005 |
| | **QMVC** | −0.355 | 0.038 | −0.018 | 0.918 |

6MWD; six minute walk distance, QMVC; Quadriceps maximal voluntary contraction numbers of individuals of each gender at each timepoint are given in brackets.

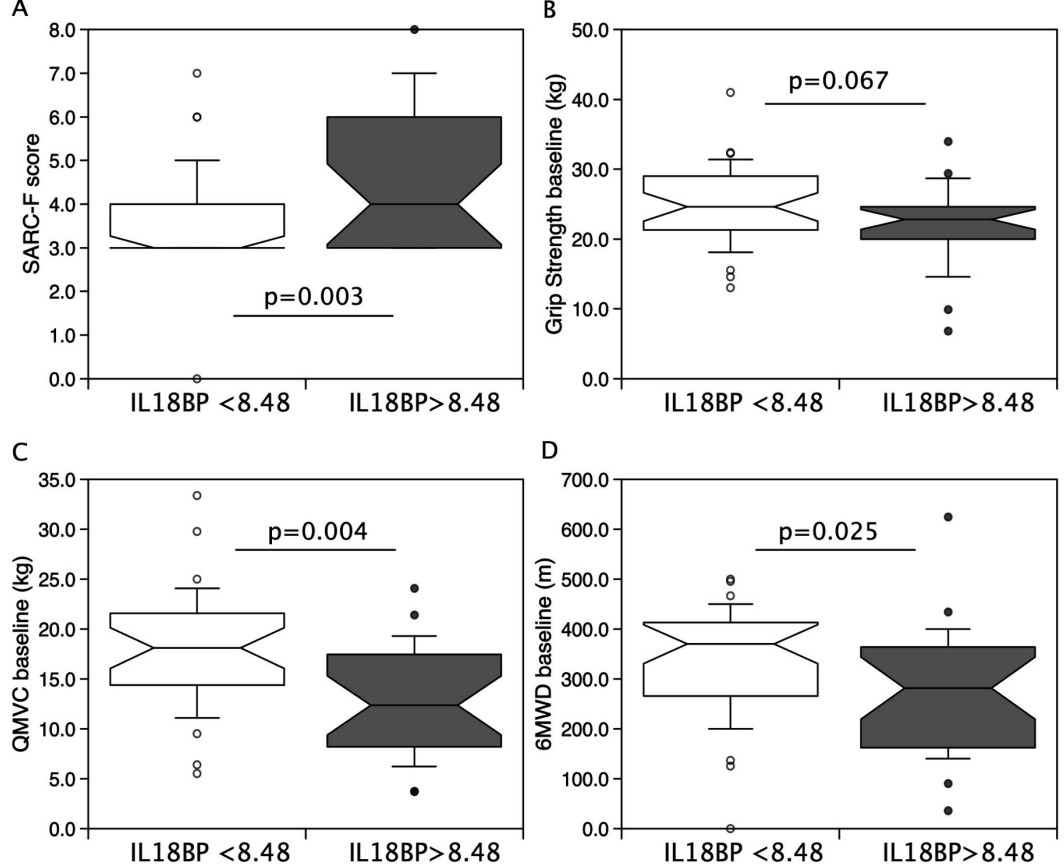

**Fig 3. Individuals with IL-18BP above median have a higher SARC-F score and poorer physical performance.** IL-18BP was measured in the plasma of individuals recruited to the LACE trial and median IL-18BP was determined as 8.48ng/mL. Men were divided into those with IL-18BP<8.48ng/mL and IL-18BP>median. SARC-F score **(A)** was higher in those with IL-18BP>8.48ng/mL. Median grip-strength **(B)**, QMVC **(C)** and 6MWD **(D)** were all lower in those with IL-18BP>8.48ng/mL although this did not reach statistical significance for grip strength. (p values calculated by Mann Whitney U test).

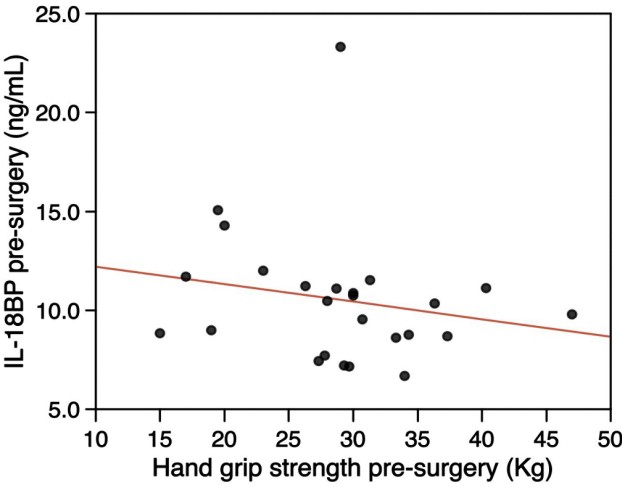

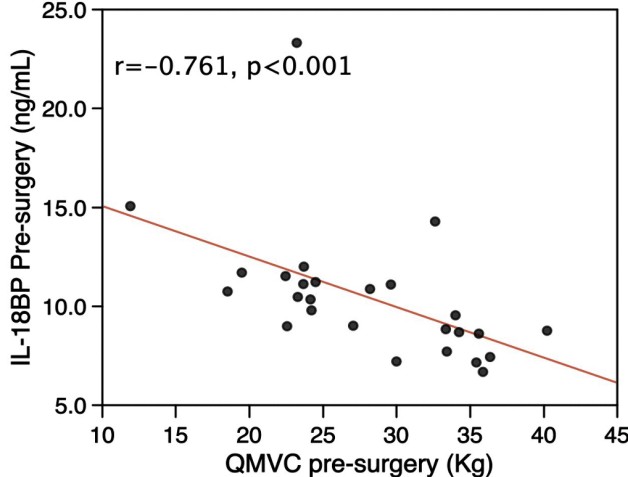

**Fig 4. Pre-surgical IL-18BP is inversely associated with strength.** Plasma IL-18BP quantified by ELISA was inversely proportional to grip-strength **(A)** and QMVC **(B)** within the whole cohort. The association with QMVC but not grip-strength was retained when men were analysed as a group. Correlations were determined by Spearman's rank.

**Table 3. Gene Set enrichment analysis for muscle genes ranked by association with IL18BP prior to surgery.**

| Hallmark Gene set | NES | NOM p-val | FDR q-val |
|---|---|---|---|
| **OXIDATIVE PHOSPHORYLATION** | −1.946 | 0.006 | 0.046 |
| **MTORC1 SIGNALING** | −1.311 | 0.138 | 0.476 |
| **INTERFERON GAMMA RESPONSE** | −1.194 | 0.240 | 0.488 |
| **COMPLEMENT** | −1.175 | 0.262 | 0.390 |
| **APICAL JUNCTION** | −1.140 | 0.289 | 0.354 |
| **XENOBIOTIC METABOLISM** | −0.649 | 0.895 | 0.888 |

NES; net enrichment score, NOM p-val; nominal p-value, FDR q-val; False discovery rate (q-value).

p=0.0014). The most tightly correlated mitochondrially associated gene was TRMT10C a subunit of the mitochondrial RNAseP (r=−0.71, p=0.002). Other genes in this group included COX20, COQ3, UQCRB, SDHAF3 and NDUFA4, all components of the electron transport chain or encode proteins required for its formation.

IL-18BP binds both IL-18 and IL-37 making it unclear which (if either) protein is the likely contributor to muscle strength. We therefore quantified IL-18 and IL-37 in both cohorts to determine whether either protein was responsible for the associations of IL-18BP with muscle strength. Of note, IL-18 was not reported in the SOMAscan analysis. IL-37 measured by ELISA did not match IL-37 measured by SOMAscan, however, given the use of the IL-37 ELISA in other publications, the ability to measure IL-37 in a larger number of samples and the fact that our median IL-37 measured by the ELISA (66.6 pg/mL) was in the range reported elsewhere (see [32] for meta-analysis) we only report comparison of the IL-37 ELISA data with strength.

In the LACE cohort, total IL-18 did not associate with muscle strength in men or women measured either as hand grip strength or as QMVC, nor did it associate with 6MWD (Table 4). Calculated free IL-18 also showed no association with strength in either gender but was proportional to 6MWD in men (r=0.397, p=0.004, Table 4). IL-37 levels were very variable. The majority of individuals had low levels of plasma IL-37, but some had much higher plasma IL-37. This pattern was similar to that we have seen for IL-6. We therefore compared IL-37 with IL-6 in the same patients and showed that IL-37 was directly proportional to IL-6 and TNFα (r=0.487, p<0.001). In this cohort IL-37 did not associate with any physiological parameter in either gender (Table 4).

In the acute sarcopenia cohort, while plasma IL-18 did not change in response to surgery, plasma IL-37 was significantly increased in post-surgery plasma samples compared to the pre-surgery samples (Fig S3 in S4 File). Prior to surgery, neither total IL-18 nor free IL-18 associated with in muscle mass or strength at the same time point (Table S9 in S1 File). Neither IL-18 nor free IL-18 measured in the pre-surgery samples associated with loss of strength or muscle cross sectional area over the 7 days (Table S10 in S1 File).

Pre-surgery plasma IL-37 was negatively associated with QMVC (r=−0.334, p=0.034) and muscle cross sectional area (r=−0.384, p=0.015, Table S9 in S1 File) but post-surgery plasma IL-37 did not show any association with loss of muscle strength (Table S10 in S1 File).

## Discussion

In this study, we identified IL-18BP as a protein elevated in response to surgery and associated with better retention of hand grip-strength 7 days after surgery. This observation led us to hypothesise that higher levels of IL-18BP would be positively associated with strength and physical performance in individuals with sarcopenia. However, comparison of

**Table 4. Total IL18, free IL18 and IL37 in sarcopenic individuals.**

|  |  | Women | | Men | |
| --- | --- | --- | --- | --- | --- |
|  | **Parameter** | $r_s$ | p | $r_s$ | p |
| **Baseline parameter vs baseline total IL18** | **Grip strength** | −0.169 | 0.169 | −0.179 | 0.189 |
|  | **6MWD** | −0.211 | 0.087 | 0.213 | 0.118 |
|  | **QMVC** | −0.149 | 0.240 | −0.017 | 0.904 |
| **Baseline parameter vs baseline free IL18** | **Grip strength** | −0.120 | 0.346 | 0.094 | 0.502 |
|  | **6MWD** | −0.136 | 0.289 | 0.397 | 0.004 |
|  | **QMVC** | −0.009 | 0.486 | 0.207 | 0.160 |
| **baseline parameter vs total IL37** | **Grip strength** | 0.057 | 0.644 | 0.212 | 0.119 |
|  | **6MWD** | 0.180 | 0.144 | −0.060 | 0.660 |
|  | **QMVC** | 0.080 | 0.530 | −0.225 | 0.116 |

6MWD; six minute walk distance, QMVC; quadriceps maximal voluntary contraction.

plasma IL-18BP in sarcopenic individuals showed the opposite associations with lower IL-18BP associated with higher strength measured both as grip-strength and as QMVC, as well as greater 6MWD. Comparison of baseline IL-18BP levels in patients undergoing surgery also showed an association of lower baseline IL-18BP with greater strength. Comparison of IL-18BP with the muscle transcriptome from the same patients showed an inverse association with genes associated with mitochondrial oxidative phosphorylation. As IL-18BP binds both IL-18 and IL-37, we then quantified these proteins in the same sample sets. In the sarcopenic population, free IL-18 was positively associated with 6MWD in men, but did not associate with muscle strength, whereas IL-37 did not associate with any of the physiological parameters measured. In the surgery cohort, there was no association of free IL-18 with muscle mass or strength. The observation that higher IL-18BP associates with the retention of strength in the acute condition but with poorer strength and performance in sarcopenia and under baseline conditions in patients about to undergo surgery, suggest that the effects of the IL-18/IL-37/IL-18BP system is context dependent, so these two situations are discussed separately.

### The response to surgery

The interaction between IL-18BP and IL-18 is thought to be inhibitory with IL-18BP sequestering IL-18, thereby restricting the activity of this member of the IL-1 family of pro-inflammatory cytokines [33,34]. Pro-inflammatory cytokines can promote muscle loss and reduce muscle strength. The simplest explanation of the association of post-surgery IL-18BP with the retention of muscle strength in patients therefore is that in those with the highest IL-18BP have the lowest level of IL-18 activity, where IL-18 is a significant contributor to the loss of muscle strength. In support of this argument, skeletal muscle is known to have IL-18 receptors and blocking IL-18 with a neutralising antibody inhibits HMGB1 induced atrophy in C2C12 myotubes [35]. However, the lack of change in circulating IL-18 levels in response to surgery argues against an independent role for IL-18 in the loss of strength after surgery. Therefore, if IL-18 is contributing to the loss of strength, it will be by acting synergistically with inflammatory cytokines that do increase in response to surgery (e.g., IL-6, IL-24). Alternatively, IL-18BP could be acting as a surrogate measure of the natural inactivation of the inflammatory response to surgery, where those who do this fastest or to the greatest extent lose least strength. It is known that inactivating inflammatory responses is critical, and that IL-18BP is involved to some extent as reduced inactivation of IL-18 by IL-18BP is a key component of macrophage activation syndrome [20].

IL-37 is a known anti-inflammatory cytokine that binds to the IL-18α receptor (IL-18Rα) and recruits the orphan receptor IL-1R8 to inhibit inflammatory mediators including NF-kB and mitogen activated protein kinases. The anti-inflammatory activity of IL-37 appears to be biphasic with anti-inflammatory effects reduced at high concentration by at least two mechanisms – firstly, by multimerization [36] and secondly by binding to IL-18BP and reducing the ability of IL-18BP to neutralise IL-18 function complicating the analysis of IL-37 function in humans. IL-37 has been shown to be increased in response to inflammation [37] and consistent with this we found that IL-37 was increased in response to surgery, raising the possibility that pro-inflammatory effects of IL-18 become suppressed in these conditions. It is therefore difficult to draw conclusions about the mechanistic significance of any associations of IL-37 with strength.

### Under normal physiological conditions

The inverse association of IL-18BP with strength in both the sarcopenic population and in those about to undergo surgery runs counter to a role for an IL-18 induced inflammatory response reducing muscle strength. The observation is also contrary to analyses of the associations of IL-18 with physical performance in multiple previous studies. For example, in Japanese men total IL-18 was negatively correlated with daily physical activity [38] and in a murine model endurance training repressed IL-18 expression [39]. Furthermore, other studies have shown increased levels of circulating IL-18 in sarcopenic individuals [22]. Our measurements of total IL-18 did not show a similar trend and our analysis of free IL-18 suggested the opposite effect. It is possible that the difference is one of population selection. We did not have a "control population" as all participants in the LACE study were sarcopenic and this may have influenced the outcome. The previous study also

combined males and females, whereas we analysed the genders separately (though combining them does not change our findings). Whilst both studies used the same kit for analysing IL-18, our IL-18 levels were lower than those reported in the other study. However, it should be noted that these studies only measure total IL-18; IL-18BP was not measured so it is not possible to know what the free IL-18 concentration was and therefore whether physical performance was associated with increased bioavailable IL-18.

Given the consistency of the observation of an inverse association of IL-18BP with strength (identified in both our sarcopenic population and a population of individuals about to undergo surgery), it seems unlikely that it occurred by chance. Maintenance of mitochondrial function or a greater mitochondrial number is associated with greater physical performance and with the preservation of muscle mass, with mitochondrial dysfunction identified as a component of muscle loss and poor physical performance in old age [40], chronic disease [41,42] and in patients undergoing surgery [19]. Consistent with increased IL-18BP associating with reduced mitochondrial content, we found that IL-18BP in plasma was inversely associated with gene sets associated with mitochondria and mitochondrial function. The genes showing these associations included components of the electron transport chain (including NDUFA4, SDHAF3, UQCRB and COX20; components of complex I, II, III and IV respectively), genes required for the synthesis of the electron transport chain (including COQ3; required for coenzyme Q synthesis) as well as genes involved in control of mitochondrially encoded RNA processing (TRMT10C). These factors fit with the inverse association of IL-18BP with mitochondrial gene expression and with better physical performance in our cohorts. There are several explanations for these findings. The first is that IL-18 has effects under normal/low inflammatory conditions that promote muscle homeostasis and oxidative metabolism. This possibility is supported by the observation that IL-18R$^{-/-}$ mice are heavier than their wild-type counterparts with increased adiposity and increased insulin resistance [43]. The IL-18R$^{-/-}$ mice store lipid in their muscle and this appears to be a consequence of a failure to activate AMPK in the skeletal muscle and other metabolic tissues which reduces fat oxidation [17]. Activation of AMPK is a known pathway to increase PGC1-alpha activity and thereby mitochondriogenesis [44] so would have the effect we observe. Furthermore, low levels of IL-18 and IL-18R have been shown in patients with HIV-associated muscle lipid dystrophy, again raising the possibility of IL-18 promoting skeletal muscle mitochondriogenesis and the use of lipid oxidation as a fuel [23]. Our finding of a positive association of free IL-18 with 6MWD in men is consistent with such a basal role for IL-18. However, these observations are counter to those showing elevated IL-18 in obese patients [45], but it is possible that these increased levels are a reflection of a resistance to IL-18 signalling [46].

The second possibility is that the association of IL-37 with IL-18BP is key to the associations of IL-18BP with strength under basal conditions. The anti-inflammatory properties of IL-37 would mean that binding of IL-37 by IL-18BP could increase sensitivity of an individual to inflammation and thereby contribute to greater protein turnover and muscle loss/weakness [47]. In addition to its anti-inflammatory effects, IL-37 modifies metabolism and increases endurance. Indeed, administration of IL-37 to either young healthy or older mice increased running endurance [48,49]. This activity requires the decoy IL1R8 receptor and leads to an increase in AMPK activation. As activation of AMPK promotes mitochondriogenesis, such an effect would be consistent with the negative association we see between IL-18BP, physical performance in both cohorts and the mitochondrial gene expression in the acute sarcopenia cohort. Our analysis of IL-37 would tend to argue against this possibility, but the sensitivity of IL-37 levels to inflammatory stimulation and the complexities of IL-37 biology described above mean that a role for IL-37 in this effect cannot be ruled out. A third possibility is that the associations of IL-18BP with strength are independent of IL-18/IL-37 either because IL18-BP is acting by an, as yet unknown mechanism or because it is reflective of a greater response to switch off the inflammatory system by one or more mechanisms rather than having a direct effect.

Our findings on the association with strength in sarcopenia are limited to men as no similar associations were seen in the women in the study. The mechanisms contributing to this observation are not clear but may reflect differences in the fibre type proportions in men and women. Women have a higher proportion of type I fibres than men and these fibres are richer in mitochondria. Consequently, the effects of IL-18BP on mitochondrial gene expression may either be fibre specific or insufficient to demonstrate an association with physiological function in women (i.e., there is a ceiling effect in women).

## Limitations of the study

The main limitations of this study are that we cannot show direct effects of IL-18BP on skeletal muscle metabolism nor verify which IL-18BP ligand is responsible for the apparent effects of IL-18BP. However, IL-18BP was not the only component of the IL-18 signalling pathway that associated with grip strength in the SOMAscan data set suggesting that aspects of IL-18 signalling are important in the loss of muscle strength that occurs after surgery. A second limitation is the sample size with the surgery cohort limited to 30 SOMAscan assay and the sarcopenia study limited to 129 individuals for the ELISA analysis. The number for which we had both transcriptomic and proteomic data was particularly limited making it possible that the associations with mitochondrial gene expression are serendipitous. However, mitochondrial gene expression is a known correlate with exercise performance and both IL-18 and IL-37 have been shown to modify pathways associated with mitochondrial gene expression. Thus, our data have an internal consistency strongly suggestive of mitochondria being a major link and are consistent with other studies. One strength of our data is that we have identified the same association (IL-18BP inversely associating with strength) in two completely different cohorts, those about to undergo surgery and older individuals with sarcopenia. Our original data (inverse association of IL-18BP following surgery with loss of strength) indicated that IL-18BP would help to maintain strength and is consistent with other data looking at levels of IL-18 in disease and associated physiological function. Whether the difference between the basal data sets and the data following a large inflammatory response reflects different aspects of IL-18 function or different ligands remains to be determined. One final strength of our study is that we quantified IL-18BP by two different methods in one set of samples and these measurements were in good agreement.

The SOMAscan study identified other proteins that associated with loss of strength that are not components of the IL-18 signalling system, but which have not been followed up either in the acute sarcopenia cohort or the LACE cohort. At least some of these proteins are also likely to contribute to the loss of muscle strength following surgery.

## Future directions

Identification of the mechanisms leading to the different associations of IL-18BP that we identify in this study would require administration of either or both of IL-18 and IL-18BP or an anti-IL-18 antibody. Phase II clinical trials of IL-18BP are underway in renal transplantation, Crohn's disease and Behcet's disease. These trials suggest that a clinical trial of IL-18BP in response to surgery would be practicable and could demonstrate preserved strength, comparison of this effect with that of an anti-IL-18 antibody would demonstrate that whether binding of IL-18 was sufficient to preserve strength in these patients. Determination of the mechanism by which IL-18BP reduces strength or mitochondrial function in humans is more complex. As described above data derived from mice are consistent with such an effect but experiments in humans would require prolonged treatment of patients with reduced exercise performance with very low dose IL-18 or a neutralising antibody to IL-18BP. Given the contribution of IL-18 to MACS and the protective role IL-18BP plays in this syndrome such an experiment would be unfeasible.

## Conclusion

In conclusion, we have found that IL-18BP associates negatively with physical performance under basal conditions in patients about to undergo surgery and in older individuals with sarcopenia. Conversely, following surgery the elevated levels of IL-18BP are associated with the retention of strength. The mechanism by which these different effects of IL-18BP occur requires further investigation.

## Supporting information

**S1 File. Supplementary Tables S1, S2, S8–S10.**
(DOCX)

**S2 File. Supplementary Tables S3–S6.**
(XLSX)

**S3 File. Supplementary Table S7.**
(XLSX)

**S4 File. Supplementary Figures S1–S3; Supplementary Figure Legends.**
(ZIP)

## Author contributions

**Conceptualization:** Aaron C. Hinken, David Neil, Alan Russell, Miles D. Witham, Mark J. Griffiths, Paul R. Kemp.

**Data curation:** Paul R. Kemp.

**Formal analysis:** Paul R. Kemp.

**Investigation:** Richard Paul, Christos Rossios, Mark J. Griffiths.

**Project administration:** Miles D. Witham, Paul R. Kemp.

**Supervision:** Paul R. Kemp.

**Writing – original draft:** Paul R. Kemp.

**Writing – review & editing:** Aaron C. Hinken, David Neil, Alan Russell, Miles D. Witham, Mark J. Griffiths.

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
