## [Decision Letter · Decision Letter 0]

21 Jun 2025

Dear Dr. Kemp,

Thank you for submitting your manuscript to PLOS ONE. After careful consideration, we feel that it has merit but does not fully meet PLOS ONE’s publication criteria as it currently stands. Therefore, we invite you to submit a revised version of the manuscript that addresses the points raised during the review process.

Please submit your revised manuscript by Aug 05 2025 11:59PM. If you will need more time than this to complete your revisions, please reply to this message or contact the journal office at plosone@plos.org . A rebuttal letter that responds to each point raised by the academic editor and reviewer(s). You should upload this letter as a separate file labeled 'Response to Reviewers'.A marked-up copy of your manuscript that highlights changes made to the original version. You should upload this as a separate file labeled 'Revised Manuscript with Track Changes'.An unmarked version of your revised paper without tracked changes. You should upload this as a separate file labeled 'Manuscript'.

We look forward to receiving your revised manuscript.

Kind regards,

Karthikeyan Thiyagarajan, PhD

Academic Editor

PLOS ONE

**Journal Requirements:**

1. When submitting your revision, we need you to address these additional requirements. Please ensure that your manuscript meets PLOS ONE's style requirements, including those for file naming. The PLOS ONE style templates can be found at https://journals.plos.org/plosone/s/file?id=wjVg/PLOSOne_formatting_sample_main_body.pdf and https://journals.plos.org/plosone/s/file?id=ba62/PLOSOne_formatting_sample_title_authors_affiliations.pdf 2. We note that there is identifying data in the Supporting Information file “SOMAscan and RNAseq data for IL18 paper”. Due to the inclusion of these potentially identifying data, we have removed this file from your file inventory. Prior to sharing human research participant data, authors should consult with an ethics committee to ensure data are shared in accordance with participant consent and all applicable local laws. Data sharing should never compromise participant privacy. It is therefore not appropriate to publicly share personally identifiable data on human research participants. The following are examples of data that should not be shared: -Name, initials, physical address-Ages more specific than whole numbers-Internet protocol (IP) address-Specific dates (birth dates, death dates, examination dates, etc.)-Contact information such as phone number or email address-Location data-ID numbers that seem specific (long numbers, include initials, titled “Hospital ID”) rather than random (small numbers in numerical order) Data that are not directly identifying may also be inappropriate to share, as in combination they can become identifying. For example, data collected from a small group of participants, vulnerable populations, or private groups should not be shared if they involve indirect identifiers (such as sex, ethnicity, location, etc.) that may risk the identification of study participants. Additional guidance on preparing raw data for publication can be found in our Data Policy (https://journals.plos.org/plosone/s/data-availability#loc-human-research-participant-data-and-other-sensitive-data) and in the following article: http://www.bmj.com/content/340/bmj.c181.long. Please remove or anonymize all personal information (<specific identifying information in file to be removed>), ensure that the data shared are in accordance with participant consent, and re-upload a fully anonymized data set. Please note that spreadsheet columns with personal information must be removed and not hidden as all hidden columns will appear in the published file. 3. Please remove all personal information, ensure that the data shared are in accordance with participant consent, and re-upload a fully anonymized data set.  Note: spreadsheet columns with personal information must be removed and not hidden as all hidden columns will appear in the published file. Additional guidance on preparing raw data for publication can be found in our Data Policy (https://journals.plos.org/plosone/s/data-availability#loc-human-research-participant-data-and-other-sensitive-data) and in the following article: http://www.bmj.com/content/340/bmj.c181.long.

**Additional Editor Comments:**

Dear Authors,

I appreciate your research on the immunological aspects in sarcopenia, looking at how proinflammatory interleukin-18 is linked to age-related muscle degeneration issues, and you have shown that the soluble IL-18 binding protein neutralization activity on IL-18, which in turn lowers inflammation-related consequences. The study further compared the IL-18bp and negative regulation of specific mitochondrial genes, which was revealed through the transcriptome analysis. But here, perhaps you can talk about the negative enrichment of specific mitochondrial genes rather than directly saying it about the mitochondrial genes (as it is like globally in mitochondria). Most of the positive correlations between the parameters and IL-18bp are weak to very weak, and there are even negative correlations among the sarcopenia patients. Why didn't you take into consideration healthy individuals? Because it is always better to compare the normal individuals as well (also, you have mentioned we did not have a “control population”). Besides the most negative correlations, the SARC-F score was positively correlated with men and had a weak positive correlation in women, and why the SARC-F follow-up was not assessed after 6 months and 12 months is not shown in Table 2. It would have been better if you could have added the regression lines of the correlations from the correlation figures. Thus, the IL-18BP with the SARC-F score for men could be a potential marker for sarcopenia due to positive correlation.

I have some additional suggestions and comments

You have mentioned that the surgery increased and decreased the concentrations of 295 and 301 proteins, respectively; the IL-18bp is the one among them, and it is a potential factor against the loss of strength from the muscles of the sarcopenic patients. Could you please slightly explain about the other proteins if there is information? Besides the cells of the immune system and muscle cells, do you expect any other non-haematopoietic cells to express these proteins? Is there a contribution of cells from other tissues concerning these proteins? Does the SOMAscan identify the proteins from specific tissues? Could you please provide the source link for the SOMAscan?

Lack of balance between the IL-18 and IL-18bp is the main factor behind sarcopenia with a loss of muscle mass and strength. Besides, did you compare this factor with any other diseases that are directly or indirectly related to sarcopenia?

The elevation of IL-18 with individuals with sedentary lifestyles and old populations is generally observed. Also, a similar finding was done before with IL-12, also for the association with sarcopenia (https://pmc.ncbi.nlm.nih.gov/articles/PMC8140914/), so how do you compare other interleukins? Do you consider IL-18 the major contributor cytokine to sarcopenia? Perhaps the expression of the proinflammatory factors commonly triggered by a sedentary lifestyle, etc., and maybe more gene networks and other transcription factors are behind it. So how do you infer this?

Please mention the expansion for the abbreviations at least one time in the introduction or elsewhere; for instance, please mention the expansion of LACE concerning your study (Leucine or Angiotensin Converting Enzyme inhibitors for sarcopenia). There are other expansions also: the Laparoscopic Approach to Cancer of the Endometrium (LACE) trial. So a reader who is not an expert in sarcopenia may not immediately recognise the term unless you mention it. So please mention the expansion.

There must be proinflammatory inhibitor drugs available for those who have undergone aortic surgery, and these drugs may act like IL-18bp to neutralize IL-18, so do you have an idea to compare them with your study now or with your future studies?

Achison et al., 2020 (https://pmc.ncbi.nlm.nih.gov/articles/PMC8977979/) found there is no efficacy of ACE inhibitors or leucine for the muscle mass or any improvement associated with sarcopenia, so why did you choose? Is there any report shows the association prior to Achison et al., 2020 et al., study?

Gene enrichment negative associations with circulating IL-18BP show the negative association with genes involved in different metabolic pathways, including oxidative phosphorylation, IFN-gamma response, etc.; however, it didn't show any specific genes with negative enrichment besides the pathway. It would be better to describe the specific genes from the gene enrichment analysis.

How did you choose 8.48 ng/mL of plasma IL-18bp as the median point for comparing the ARC-F score, QMVC, etc.? Whether any previous study standardized the specific point (8.48 ng/mL) of the measurements or the median point is based on your study/trial only; for instance, you have mentioned "SARC-F score (A) was higher in those with IL-18BP > 8.48 ng/mL" in accordance with Figure 3.

Seems the Figure 3 with IL-18BP < 8.48 having some skewed distribution, have you normalized the data using log or square root transformation prior to the analysis concerning the plasma IL-18BP data?

Your study shows IL-18 has no association with muscle strength as expected, as it acts as a proinflammatory factor; however, 6MWD (walking) shows a positive correlation with IL-18 in men, not in women. It means physical activity (walking) is inducing the proinflammatory cytokine (IL-8) in man? Please clarify.

IL-18 is also associated with several autoimmune diseases, and there are reports showing the isoforms of IL-18bp (https://pmc.ncbi.nlm.nih.gov/articles/PMC15564/), so referring to these kinds of studies may provide a better clue. IL-18 is known to induce IFN-γ production there by inducing the Th1 cells. So, it is better to describe how the IL-18 specifically acts on muscle cells with sarcopenia, thereby reducing the mass of the tissues and strength.

Your article is written well; however, it needs to be more precise. Please submit the RNA data set to NCBI. Have you compared the Gene Expression Omnibus? It may provide additional evidence of the gene expression, especially when dealing with mitochondrial genes. So please address the comments of the reviewers and mine, and revise the manuscript accordingly.

Reviewers' comments:

Reviewer's Responses to Questions

**Comments to the Author**

1. Is the manuscript technically sound, and do the data support the conclusions?

Reviewer #1: Yes

Reviewer #2: Partly

2. Has the statistical analysis been performed appropriately and rigorously?

Reviewer #1: Yes

Reviewer #2: Yes

3. Have the authors made all data underlying the findings in their manuscript fully available?

Reviewer #1: Yes

Reviewer #2: Yes

4. Is the manuscript presented in an intelligible fashion and written in standard English?

Reviewer #1: Yes

Reviewer #2: Yes

**Reviewer #1:** Thank you for submitting your manuscript entitled "IL-18 BP, a biomarker of strength maintenance after surgery but reduced physical performance in age - related sarcopenia" to our journal. The combination of proteomic screening and subsequent validation in two different cohorts provides a comprehensive approach to understanding the role of IL-18BP.

Weaknesses and Suggested Revisions:

1. The introduction could benefit from a more detailed discussion of the existing literature on IL-18BP and its ligands in the context of muscle wasting. This would help to better justify the hypothesis and provide a clearer rationale for the study.

2. The description of the SOMAscan assay and ELISA analysis lacks sufficient comprehensiveness and is overly concise.

3. In the RNAseq analysis section, it would be beneficial to specify the number of genes analyzed and the criteria used for filtering genes before correlation analysis.

4. The discussion touches on the potential mechanisms underlying the associations observed, but it could be expanded to more thoroughly explore the biological plausibility of the findings.

**Reviewer #2:** Summary:

This study primarily aimed at identifying biomarkers of acute and chronic sarcopenia, a condition characterized by loss of muscle mass and/or function. The authors began by examining changes in plasma proteins in relation to muscle mass or function loss in a cohort of patients who underwent aortic surgery, which is a well-accepted model for acute sarcopenia. They report approximately 600 proteins being up- or downregulated 24 hours post-surgery. Through correlation analyses, they uncovered IL-18 binding protein (IL-18BP) as being positively associated with muscle function preservation in the grip strength test performed 7 days post-surgery. To ascertain the validity of those results, they performed ELISA assays to quantify IL-18BP as well as IL-18. To determine whether IL-18BP may also serve as a biomarker for chronic sarcopenia, they revisited a previously generated dataset from a separate cohort of patients affected with chronic sarcopenia to determine whether there was a similar correlation between IL-18BP and functional data. They found that, unlike in the aortic surgery patients (acute sarcopenia), IL-18BP levels were negatively associated with muscle function.

Strengths:

Sarcopenia is a generalized disorder characterized by a significant loss in skeletal muscle mass and/or function, which leads to increased risk of noncommunicable diseases and mortality. With the sedentary lifestyle crisis, sarcopenia is a significant contributor to the global burden of disease, and understanding the mechanisms driving this process or identifying biomarkers that could predict the onset or the progression is extremely relevant to human health.

Biomarkers that do not require invasive techniques are ideal for a translational approach, especially when linked to functional outcomes that can be employed in the clinic.

The rationale for the study was well presented. The hypothesis was clearly stated. The techniques and statistical approaches were appropriate and rigorous.

Potential areas for development:

While these results are extremely relevant to the field, the manuscript is currently lacking a concise overview of the mechanisms contributing to “acute” versus “chronic” sarcopenia. Recovery from invasive surgery and chronic sarcopenia, such as during aging, are distinct physiological states differentially impacting muscle structure and function. While inflammation is an unequivocal contributor to both conditions, it is also important to consider that the effects of pro-inflammatory cytokines are heavily dependent on the timing, location, and duration of stimuli. From an Exercise Physiologist’s standpoint, inflammation is a crucial response facilitating muscle injury repair. In fact, transient elevation of pro-inflammatory cytokines is a normal response to exercise. A good example is IL-6, a pro-inflammatory cytokine that has been categorized as a top “Exerkine” and a “Myokine”. Upon acute injury or resistance exercise, skeletal muscle produces IL-6 to activate satellite cells (muscle stem cells). This is a fundamental mechanism for muscle recovery, and this is why inflammation is essential after exercise training or acute muscle injury. What may cause a lot of confusion in the scientific community is the fact that, while the transient release of pro-inflammatory factors is beneficial, chronic elevation of these factors becomes pathologic and can cause the opposite effects. (See: https://www.sciencedirect.com/science/article/pii/S155041310700366X and https://pmc.ncbi.nlm.nih.gov/articles/PMC8306542/).

Given the contrasting results presented in the paper between the acute versus chronic models of sarcopenia, it is important to discuss these concepts. This may also help with the interpretation of the results. Including one paragraph on the role of inflammation in skeletal muscle physiology would be very helpful.

The authors found very interesting results on IL-18/IL-18BP in these 2 cohorts of patients. IL-18 signaling in skeletal muscle has documented effects that are not reported or discussed in the paper such as induction of AMPk. A more thorough review of literature on this topic is highly recommended. Additional information on IL-18 may be included after the additional paragraph on inflammation and skeletal muscle.

Here are some references to look at:

https://pmc.ncbi.nlm.nih.gov/articles/PMC3749341/

https://www.frontiersin.org/journals/endocrinology/articles/10.3389/fendo.2022.971745/full

There is also a need to improve clarity and conciseness. I provided several examples below, but there are more, and I would encourage a senior co-author or collaborator to proofread before resubmission.

General minor comments:

1) The term gender should be avoided and updated to sex. (Using the find function, I found 4 occurrences: p. 9, 11 (2X), and 21.

2) Please, use the same terminology throughout the manuscript (Aortic surgery or Acute Sarcopenia). The term “acute sarcopenia” is appropriate and may contribute to better manuscript cohesion.

3) There are several double negative statements, which can become difficult to understand. An example is provided below. The authors should review the entire manuscript with this vision.

4) Data generated from a previously published clinical trial were analyzed from a different perspective. Therefore, terms like “re-examined” or “re-analyzed” could be updated to “examined” and “analyzed”… a previously published dataset.

Specific Comments:

Abstract:

P.2. – Line 11: Please clarify what the LACE trial is at the first occurrence.

P.2. – Line 4-5: Clarity: “We used a proteomic screen of plasma to identify proteins associated with strength loss.” Since you are also reporting a positive association with skeletal muscle function, the statement should be updated to “we used a proteomic screen of plasma to identify proteins associated with changes in strength”.

P.2 – Line 8-9: Avoid speculation: “Analysis of the day 1 protein levels identified IL-18BP as a potential protective factor against loss of strength.”

The term “protective” may be speculative. I would recommend updating to “associated”. Also, two negatives in a sentence can become confusing (against, loss). Perhaps, rephrasing to something like: “Analysis of the day 1 protein levels identified IL-18BP as being positively correlated with the maintenance of grip strength”.

P.2. – Line 19: Contrasting wording: “negative enrichment for mitochondrial genes”. It is unclear what this means. Are you referring to DEGs (Differentially expressed genes)? If so, please use universal bioinformatic terminology (Up/Down; or Upregulated/Downregulated).

Introduction:

I gave general recommendations in the “Potential areas for development” section above. I will provide specific feedback in the revised version of the manuscript.

Methods:

P5. Line 11-12: Please rephrase this statement and specify the muscle loss threshold at which patients were classified as sarcopenic. Alternatively, you may want to simply update “muscle loss” to “muscle cross-sectional area”.

Results:

P.8- Line 17-18: Citation: “A more complete analysis of these changes will be described

Elsewhere.” Did you mean that these were previously published? Unless a reference can be provided, this statement should be removed since it’s in the future.

P.9 Line 5- “Human surgical model” should be updated to “Human patient population” or “cohort of patients”.

P.9 – Line 10-15: Clarity: I had to re-read to notice the difference between the first and second sentences. Beginning sentences by stating the time point would help with clarity.

P.9- Line 8-22: Clarity: Given that males and females were analyzed separately, including results from both sexes in a single sentence can become confusing. I would recommend presenting the results in separate sentences and avoiding comparing the results. For example: update “In men but not women,” to “In men,”.

P. 10 – Line 7-10: This sentence is very difficult to understand and unclear. Negatively associated with what? Perhaps breaking down this sentence into 2 sentences would help with clarity.

P. 10 – Line 17: Contrasting wording: “negatively enriched”

P.10 – Line 22-24: Terminology: Since the data reported is limited to basic muscle function tests and imaging, the term physiology is too broad. The words “morphology and function” would be more appropriate.

Discussion:

As this section may require intensive recrafting to address my comments, I will provide additional feedback when receiving the revised manuscript.

Best of luck!

**Do you want your identity to be public for this peer review?** For information about this choice, including consent withdrawal, please see our Privacy Policy

Reviewer #1: No

Reviewer #2: No

---

## [Author Response · Author response to Decision Letter 1]

6 Aug 2025

I have included the response to reviewers as part of the cover letter so that the comments and responses can be viewed easily based on colour

---

## [Decision Letter · Decision Letter 1]

14 Oct 2025

Dear Dr. Kemp,

Thank you for submitting your manuscript to PLOS ONE. After careful consideration, we feel that it has merit but does not fully meet PLOS ONE’s publication criteria as it currently stands. Therefore, we invite you to submit a revised version of the manuscript that addresses the points raised during the review process.

We look forward to receiving your revised manuscript.

Kind regards,

Karthikeyan Thiyagarajan, PhD

Academic Editor

PLOS ONE

Journal Requirements:

Additional Editor Comments:

Dear Authors,

I appreciate your responses and revision, and I agree with your concerns concerning the restriction of the NCBI submission of your data for the moment with the paper. But I would recommend submitting your data to the public domain, like NCBI, with your future work. As you have evaluated the negative enrichment of mitochondrial genes and provided a supplementary file (please mention the supplementary file number wherever it is not mentioned), it would be interesting to discuss some of the negatively enriched genes of mitochondria. I would add the same concerns to proteins as well. One of the reviewers also provided the comments and suggestions with a minor revision. Thus, I recommend you revise your manuscript.

Reviewers' comments:

Reviewer's Responses to Questions

**Comments to the Author**

Reviewer #1: All comments have been addressed

Reviewer #3: (No Response)

2. Is the manuscript technically sound, and do the data support the conclusions?

Reviewer #1: Yes

Reviewer #3: Yes

3. Has the statistical analysis been performed appropriately and rigorously?

Reviewer #1: Yes

Reviewer #3: Yes

4. Have the authors made all data underlying the findings in their manuscript fully available?

Reviewer #1: Yes

Reviewer #3: Yes

5. Is the manuscript presented in an intelligible fashion and written in standard English?

Reviewer #1: Yes

Reviewer #3: Yes

Reviewer #1: The authors have adequately addressed my comments and I believe that this manuscript is now acceptable for publication.

Reviewer #3: IL-18 Binding Protein (IL-18BP) is increasingly recognized as a context-sensitive immunomodulatory biomarker with emerging roles in muscle physiology and aging. While the current study provides valuable insights, several methodological and interpretative considerations should be addressed to enhance the manuscript’s clarity and rigor:

Study Limitations: A thorough discussion of the study’s limitations—including sample size, extract standardization, and potential confounding variables—is essential to appropriately contextualize the findings.

Data Interpretation: The observed discrepancy in IL-18BP levels between sarcopenic patients at baseline and post-surgical time points warrants further clarification. The authors are encouraged to propose specific recommendations and outline a roadmap for resolving this inconsistency.

Title Accuracy: To ensure terminological precision, the manuscript title should reference IL-18 binding protein rather than IL-1BP.

Sincerely,

**Do you want your identity to be public for this peer review?** For information about this choice, including consent withdrawal, please see our Privacy Policy

Reviewer #1: No

Reviewer #3: **Yes:** S.M. Abtahi Froushani

---

## [Author Response · Author response to Decision Letter 2]

17 Nov 2025

Response to editor’s and Reviewers’ comments:

I appreciate your responses and revision, and I agree with your concerns concerning the restriction of the NCBI submission of your data for the moment with the paper. But I would recommend submitting your data to the public domain, like NCBI, with your future work. As you have evaluated the negative enrichment of mitochondrial genes and provided a supplementary file (please mention the supplementary file number wherever it is not mentioned), it would be interesting to discuss some of the negatively enriched genes of mitochondria. I would add the same concerns to proteins as well. One of the reviewers also provided the comments and suggestions with a minor revision. Thus, I recommend you revise your manuscript.

Thank you for agreeing that we don’t need to include the submission number for our NCBI data in this paper. However, we would like to reassure you that the data have already been submitted to NCBI (as part of the process of submitting the other paper and I have an appropriate submission number). We have included reference to the additional file where appropriate.

We have also added a comment regarding the associations between specific genes that encode mitochondrial proteins and IL-18BP to the discussion. This section reads:

Consistent with increased IL-18BP associating with reduced mitochondrial content, we found that IL-18BP in plasma was inversely associated with gene sets associated with mitochondria and mitochondrial function. The genes showing these associations included components of the electron transport chain (including NDUFA4, SDHAF3, UQCRB and COX20; components of complex I, II, III and IV respectively), genes required for the synthesis of the electron transport chain (including COQ3; required for coenzyme Q synthesis) as well as genes involved in control of mitochondrially encoded RNA processing (TRMT10C).

Given that there was no original reviewer 3 we assume that the comments are from a further reviewer due to the lack of response from one of our original reviewers. We thank them for reviewing the paper and respond to their comments below. We note that we have successfully answered reviewer 1’s comments in our rebuttal and thank them for re-reviewing the work.

Response to the comments by reviewer 3.

IL-18 Binding Protein (IL-18BP) is increasingly recognized as a context-sensitive immunomodulatory biomarker with emerging roles in muscle physiology and aging. While the current study provides valuable insights, several methodological and interpretative considerations should be addressed to enhance the manuscript’s clarity and rigor:

Study Limitations: A thorough discussion of the study’s limitations—including sample size, extract standardization, and potential confounding variables—is essential to appropriately contextualize the findings.

The section marked Study Limitations commented on the limitation of study size but predominantly with reference to the number of samples that had overlapping protein and RNA sequencing data. We have amended this to cover the study as a whole.

This section now reads

A second limitation is the sample size with the surgery cohort limited to 30 SOMAscan assay and the sarcopenia study limited to 129 individuals for the ELISA analysis. The number for which we had both transcriptomic and proteomic data was particularly limited making it possible that the associations with mitochondrial gene expression are serendipitous.

We are not clear on which aspects of “extract standardisation” the reviewer is referring to. The ELISA was performed on diluted plasma so there was no “extraction” and all samples were diluted to the same extent, the SOMAscan data was generated performed at SOMAlogic using their proprietary methods and all samples passed their QC methods.

RNA was extracted by a normal Trizol extraction method (this has now been added to the methods) and the RNA samples and libraries were QC’d at Barts Genome centre and passed their QC metrics. This has now been made explicit and the section reads

RNA was extracted from frozen muscle biopsies taken at the time of surgery (after anaesthetisation but before surgery) using TRizol as previously described [28]. RNAseq was performed at the Genome Centre, Queen Mary University of London and all samples and generated libraries passed the QC metrics used by the Genome Centre.

Data Interpretation: The observed discrepancy in IL-18BP levels between sarcopenic patients at baseline and post-surgical time points warrants further clarification. The authors are encouraged to propose specific recommendations and outline a roadmap for resolving this inconsistency.

We are unclear what discrepancy/inconsistency in IL18-BP levels the reviewer is referring to. The sarcopenic individuals (from the LACE study) did not undergo surgery. The range of IL18-BP (median 8.48pg/mL (5.76, 12.43) seen in these individuals covers the same range covered by pre-surgical levels in the patients undergoing aortic surgery (median=10.4 (8.78, 11.70). It is true that these are statistically significantly different with the surgery patients having a higher level than the LACE patients (p=0.027) though this doesn’t appear to be the difference that the reviewer is asking about as they specify post-surgery. However, given that there is a marked increase in IL-18BP following surgery compared with pre-surgery in the same individuals and we know that the surgery initiates a marked inflammatory response, and the aortic surgery patients have a significant illness associated with increased with inflammation, it seems likely that this is the most likely cause of the difference. This interpretation would also be consistent with increases in IL-18BP in sepsis where it limits the effects of IL-18 as described in the text. Other differences in the cohort include age which may also be significant but more speculative in nature as an explanation.

The difference in IL1-8BP between samples taken, before and after surgery, from the individuals undergoing surgery is, as far as we can tell, real. The samples were taken in the same way (just 24h apart from each individual over the period of about 1 year to obtain samples for the whole cohort). There were stored together at -80�C until analysed by SOMAscan and subsequently by ELISA on aliquots that were stored separately from those that went for SOMAscan analysis. In response to surgery there is a large inflammatory response with increases in many inflammatory cytokines indicating that the increase in IL-18BP is likely a consequence of this inflammatory stimulus.

Taking this together, we do not see any discrepancy in the levels of IL-18BP that could be analysed in any “roadmap”.

If on the other hand the reviewer is talking about the discrepancy in the association between IL-18BP and strength, then that is discussed in the text in the light of the effects in MACS and the mitochondrial changes in IL-18BP-/- mice. However, we have added the following as a roadmap to understanding the situation in humans.

Future directions

Identification of the mechanisms leading to the different associations of IL-18BP that we identify in this study would require administration of either or both of IL-18 and IL-18BP or an anti-IL-18 antibody. Phase II clinical trials of IL-18BP have been started examining kidney transplantation, Chron’s disease and Behcet’s disease. These trials suggest that a clinical trial of IL-18BP in response to surgery would be practicable and could demonstrate preserved strength, comparison of this effect with that of an anti-IL-18 antibody would demonstrate that whether binding of IL-18 was sufficient to preserve strength in these patients. Determination of the mechanism by which IL-18BP reduces strength or mitochondrial function in humans is more complex. As described above data derived from mice are consistent with such an effect but experiments in humans would require prolonged treatment of patients with reduced exercise performance with very low dose IL-18 or an neutralising antibody to IL-18BP. Given the contribution of IL-18 to MACS and the protective role IL-18BP plays in this syndrome such an experiment would be unfeasible.

Title Accuracy: To ensure terminological precision, the manuscript title should reference IL-18 binding protein rather than IL-1BP.

We have modified the title accordingly.

---

## [Editor Report · Decision Letter 2]

22 Dec 2025

IL-18 Binding Protein, a biomarker of strength maintenance after surgery but reduced physical performance in age-related sarcopenia

PONE-D-25-20033R2

Dear Dr. Kemp,

We’re pleased to inform you that your manuscript has been judged scientifically suitable for publication and will be formally accepted for publication once it meets all outstanding technical requirements.

Kind regards,

Karthikeyan Thiyagarajan, PhD

Academic Editor

PLOS One

Additional Editor Comments:

Dear Authors,

After careful scientific evaluations with peer reviews, I am pleased to confirm the manuscript entitled "IL-18 Binding Protein, a biomarker of strength maintenance after surgery but reduced physical performance in age-related sarcopenia" has been accepted for publication in PLOS ONE. I suggest please add the ethical statement inside the manuscript.

Kind regards,

Karthikeyan Thiyagarajan PhD

Academic Editor, PLOS ONE.
---

## [Editor Report · Acceptance letter]

PONE-D-25-20033R2

PLOS One

Dear Dr. Kemp,

I'm pleased to inform you that your manuscript has been deemed suitable for publication in PLOS One. Congratulations! Your manuscript is now being handed over to our production team.

Kind regards,

on behalf of

Dr. Karthikeyan Thiyagarajan

Academic Editor

PLOS One